# One-Stage Multilevel Surgery for Treatment of Obstructive Sleep Apnea Syndrome

**DOI:** 10.3390/jcm10214822

**Published:** 2021-10-20

**Authors:** Gabriela Bosco, Marta Morato, Nuria Pérez-Martín, Andrés Navarro, Miguel A. Racionero, Carlos O’Connor-Reina, Peter Baptista, Guillermo Plaza

**Affiliations:** 1Department of Otolaryngology Head and Neck Surgery, Hospital Universitario de Fuenlabrada, 28942 Madrid, Spain; marta37003@hotmail.com (M.M.); n.perezmartin@hotmail.com (N.P.-M.); anavarro_18@msn.com (A.N.); gplaza.hflr@salud.madrid.org (G.P.); 2Department of Otolaryngology Head and Neck Surgery, Hospital Universitario Sanitas La Zarzuela, 28942 Madrid, Spain; 3Department of Neumology, Hospital Universitario de Fuenlabrada, 28942 Madrid, Spain; miguelangel.racionero@salud.madrid.org; 4Department of Otolaryngology Head and Neck Surgery, Hospital Quironsalud Marbella, 29603 Marbella, Spain; carlos.oconnor@quironsalud.es; 5Department of Otolaryngology Head and Neck Surgery, Hospital Quironsalud Campo de Gibraltar, 11379 Palmones, Spain; 6Department of Otolaryngology Head and Neck Surgery, Clínica Universitaria de Navarra, 31008 Pamplona, Spain; peterbaptista@gmail.com

**Keywords:** obstructive sleep apnea, multilevel surgery, tongue base surgery, sleep surgery

## Abstract

We report the results of one-stage multilevel upper airway surgery for patients who could not tolerate continuous positive airway pressure (CPAP). Patients treated with multilevel surgery at a University Hospital in 2015–2019 were identified from a prospectively maintained database. The inclusion criteria were aged 18–70 years, body mass index (BMI) < 35 kg/m^2^, apnea–hypopnea index (AHI) > 20, and lingual tonsil hypertrophy grade 3 or 4. Drug-induced sleep endoscopy was performed before surgery in all patients. Multilevel surgery was performed in one stage and included expansion sphincter pharyngoplasty (ESP), coblation tongue base reduction (CTBR), and partial epiglottectomy (PE) as required. The outcome measures were postoperative AHI, time percentage oxygen saturation < 90%, and Epworth Sleepiness Scale (ESS) score. A total of 24 patients were included: median age 49.1 years, average BMI 27.26 kg/m^2^, and 90% men. Ten patients received ESP plus CTBR plus PE, eight received ESP plus CTBR, and six received ESP plus PE. The mean preoperative AHI was 33.01 at baseline and improved to 17.7 ± 13 after surgery (*p* < 0.05). The ESS score decreased from 11 ± 5.11 to 7.9 ± 4.94 (*p* < 0.05). The surgical success rate according to Sher’s criteria was 82.3%. The median follow-up was 23.3 months (range 12–36). These findings suggest that multilevel surgery is a safe and successful treatment of OSAHS.

## 1. Introduction

Multilevel surgery in a single step for treatment of obstructive sleep apnea–hypopnea syndrome (OSAHS) is being used more frequently, although its outcomes remain a matter of discussion [1]. Patients with severe OSAHS who are prescribed continuous positive airway pressure (CPAP) often refuse to use the device. As most of these patients have multilevel sites of obstruction, including the oropharynx, hypopharynx, and larynx, the best surgical treatment must be multilevel [2]. Single-stage multilevel surgery has been shown to be safe and effective. Still, patient selection is crucial, and the surgical team must consider the potential increased risk of complications related to airway collapse.

Identifying the location of obstruction of the upper airway (UA) can be challenging but is important for obtaining successful surgery results in patients with OSAHS. To facilitate the examination of the UA, drug-induced sleep endoscopy (DISE) is essential for establishing the topographic diagnosis and identifying zone(s) of obstructions and collapse in patients with OSAHS [2,3].

Multilevel surgery can involve various options to treat hypertrophy of the tongue base, a common site of obstruction in patients with severe OSAHS [4]. The transoral robotic surgery (TORS) procedure for tongue base resection in OSAHS was described by Vicini et al. [5], according to whose study it produces excellent and safe transoral access to the tongue base and epiglottis along with surgical precision and hemostasis. TORS has been reported to achieve the best outcomes compared with other options for managing tongue base hypertrophy in OSAHS patients [6]. The three-dimensional high-definition Da Vinci surgical system allows the surgeon to identify and avoid damaging the crucial structures by working carefully, step by step, using a mix of blunt and sharp dissection.

On occasion, when the robot is unavailable, other transoral surgical options are available, including coblation tongue base reduction (CTBR) described by MacKay et al. [7]. This technique involves ablating the lingual tonsils and obstructing the base of the tongue using coblation technology. While producing minimal thermal penetration into the underlying tissues, it involves gentle tissue removal compared with electrocautery or laser. Coblation technology includes irrigation, suction, and ablation/coagulation procedures and requires fewer working hands in the field [8,9]. The Robo-Cob technique is a new procedure that uses coblation to resect but not ablate tongue base hypertrophy and involves similar exposure and operative technique as described for TORS [10,11].

Our study aimed to describe our results of one-stage multilevel UA surgery using coblation for patients with OSAHS who could not tolerate CPAP and whose multilevel DISE confirmed UA collapse.

## 2. Materials and Methods

Patients affected by severe OSAHS who had been surgically treated at the Otorhinolaryngology Department of a University Hospital from November 2015 to December 2019 were identified from a prospectively maintained database. The ethics committee approved this study (No. EC920, on 2 March 2015), and all participants signed an informed consent form.

All patients underwent diagnostic sleep polysomnography (PSG) and DISE before the surgery. The inclusion criteria were ages 18–70 years, a diagnosis of moderate OSAHS as defined by an apnea–hypopnea index (AHI) 5–29 and severe if AHI > 30, lingual tonsil hypertrophy (LTH) confirmed by nasopharyngoscopy as Friedman grade 2 or more [12], and CPAP intolerance. We excluded patients who had undergone previous surgery for OSAHS, were allergic to propofol or had obesity as indicated by a body mass index (BMI) > 35 kg/m^2^.

The study protocol included a complete medical history, with the following variables: sex, BMI, cervical perimeter, tonsil grade, Friedman tongue position, Friedman stage, and preoperative and postoperative Epworth Sleepiness Scale (ESS) score. The evaluation was performed using nasopharyngoscopy with Müller’s maneuvers in the office. DISE was performed in all patients to identify the site of obstruction and the pattern of the UA collapse. Details about PSG were collected preoperatively and postoperatively and included AHI, lowest oxygen saturation (minO_2_Sat), and time percentage with SatO_2_ < 90% (TSat90).

DISE was performed systematically in the operating room with the patient in the supine position. For sedation, the patient was premedicated with 2 mg of midazolam. A 2% propofol syringe infusion pump with target-controlled infusion (TCI) with a target concentration of 2 ng/mL and, if required, a progressive increase in the dose of 0.2–0.5 ng/mL. The sedation level was monitored using the bispectral index (BIS) (BIS Quatro^®^. COVIDien Inc., Mansfield, MA, USA). When the patient was asleep and actively snoring (BIS between 70 and 50), a flexible video endoscope (TGH Endoscopia. MACHIDA ENT-30PIII, Madrid, Spain) was used to visualize the site of collapse in the UA in real time, and the images were recorded. The findings were observed for a minimum of two cycles in each segment. The VOTE classification of Kezirian et al. published in 2011 was used to express the findings [3].

Using the DISE results, an individualized surgical plan was developed for every patient. The first step was to offer nasal surgery to optimize CPAP use, but some patients refused to use CPAP and requested curative surgery. Per our protocol, as used in our institution, in selected patients with hypopharyngeal anteroposterior collapse with tongue base hypertrophy, we performed CTBR and/or partial epiglottectomy (PE), and expansion sphincter pharyngoplasty (ESP) in one stage. The head of the probe was moved over the surface of the lingual tonsil with slight pressure on the tissue, which reduce the tissue to a fluid state, allowing it to be extracted by suction (see Appendix A).

Patients were prepared and draped for surgery in the “sniffing position” (neck flexed and head extended), and the tongue base exposed was to stay silk suture in the oral tongue to deliver tongue base. We used a distending operating laryngoscope. An assistant inserted in the mouth an uplooking endoscope 30° or 45° upwards (Figure 1). The EVac 70 Xtra HP^®^ Coblation Wand (Arthrocare Corp., ENTec Division, Sunnyvale, CA, USA) was used for resection using the settings for a tonsillectomy at a power of 7 during ablation or 5 for coagulation and cold saline irrigation. The head of the probe was moved over the surface of the lingual tonsil with slight pressure on the tissue, which reduced the tissue to a fluid state, allowing it to be extracted by suction.

The resection was performed until the epiglottis could be seen, and the vallecula was completely free of lymphatic tissue. Minor bleeding was coagulated, using the coagulation mode of the Coblator system, and more severe bleeding required monopolar or bipolar electrocautery (Figure 2).

The technique also involved a partial removal of 1/3 superior of the epiglottis in patients, mainly in patients with long, flaccid, or trapdoor epiglottis that produced obstruction during DISE. Dissection through the coblation terminal was performed in a “V” shape. This protected the airway from aspiration. In addition, the lateral collapse of the epiglottis could be resolved by this procedure.

ESP was performed according to a modification of Sorrenti and Piccin [13] with the aid of a headlight as performed for bilateral tonsillectomy.

Perioperative management, including the length of stay in hospital and complications, were recorded.

Office examinations were performed at one week, four weeks, three months, six months, and 12 months after surgery and included the ESS and examination of the UA. PSG was performed between 6 and 12 months after the surgery. The success of the surgery was defined according to Sher’s criteria [14] as a postoperative AHI <15 and/or 50% reduction in preoperative AHI.

A descriptive analysis was performed to estimate the measures of central tendency (mean and median) and dispersion (range, standard deviation, and 25th to 75th percentiles) for the quantitative variables and the frequency distribution for the qualitative variables. Normality was assessed using the Shapiro–Wilk test. A *p* value <0.05 was considered significant. IBM SPSS Statistics (version 22.0; IBM Corp., Armonk, NY, USA) was used for all the analyses.

## 3. Results

We identified a cohort of 24 patients, 90% of whom were men. The demographic characteristics of the patients included in this study are summarized in Table 1. Their mean age was 49.1 ± 9 years (range 30–69) and BMI was 27.2 ± 2.7 kg/m^2^.

All patients had a diagnosis of severe OSAHS, and the mean baseline AHI measured by PSG during the preoperative evaluation was 33.01 ± 17.53 (Figure 3a). The mean minO_2_Sat was 82.4% ± 10.30%, and the mean TSat90 was 6.70 ± 10.03. ESS average previous to surgery was 11 ± 5.11 (Figure 3b).

The entire population was previously evaluated by a pulmonologist and prescribed CPAP treatment. However, four patients refused to use CPAP (16%) and requested definitive surgical treatment. The other patients did not tolerate CPAP for >3 h/night. Surgery was recommended based on DISE findings (Table 2).

Nine (37%) of the CPAP users were primarily indicated for nasal surgery as a CPAP-optimizing treatment but did not achieve good compliance with its use. Therefore, multilevel surgery was indicated after the initial nasal surgery.

Among the 24 patients included in this analysis, multilevel surgery was performed in one step including treatment of the oropharyngeal and hypopharyngeal levels of collapse. Ten patients received ESP plus CTBR plus PE, eight received ESP plus CTBR, and six received ESP plus PE.

All patients were extubated immediately after the surgery was completed. The medium length of the hospital stay was four days. Over the first 24 h, patients were monitored in the critical care unit to allow early detection of uncommon complications associated with severe OSAHS (e.g., pulmonary distress or heart failure) or with surgery (e.g., bleeding or UA edema). The common complications were pain, minor bleeding, and mild liquid dysphagia in patients who received a PE. No major complications were reported. We had no major complications, bleeding, or edema, no need for an urgent operating room. Only expected adverse events such as dysphagia resolved in the first month.

After surgery, the ESS score was 7.90 ± 4.94, and the postoperative AHI was 17.7 ± 13 (Figure 3). All patients reported subjective improvement in their sleep quality, and most reported they did not need to use CPAP. The improvements in AHI and ESS scores were significant (Figure 4). In total, 16 patients (66%) showed a reduction in AHI > 50% after surgery, and 75% of patients had an AHI value < 20 postoperatively. According to Sher’s criteria, the surgical success rate was significant in two subgroups (Table 3). Body weight did not change significantly from before to after the operation.

There is an association between Friedman and VOTE classifications in our sample of patients. Awake endoscopy determines that when there is severe lingual tonsil hypertrophy and the Friedman stage, there is a level predictor of collapse at the retrolingual level, showing a significant correlation with VOTE only when there is a severe retrolingual collapse. Friedman’s stage is more correlated with VOTE with respect to retrolingual obstruction than Müller’s maneuver.

## 4. Discussion

We conducted this prospective evaluation of 24 patients with OSAHS who underwent DISE before UA surgery to examine the results after one-stage multilevel UA surgery for patients with OSAHS who could not tolerate CPAP. In our study, multilevel surgery in one stage resulted in significant reductions in the frequency of sleep apnea, hypopnea, and daytime sleepiness in patients with severe OSAHS for whom prior attempts at conventional treatment with a medical device such as CPAP failed.

Surgery was also associated with improvements in other PSG measures, including arterial oxygen saturation measures (minO_2_Sat and SatO_2_90), partner-reported snoring, and patient-reported sleep-specific quality of life.

The statistically significant difference in the AHI was above the established minimal clinically important difference (15 events/h) [15] but less than the a priori hypothesized difference of 20 events/h [16]. When we developed our surgical plan based on DISE findings, we allowed any combination of the following surgical procedures: tonsillectomy and ESP, CTBR, and/or PE. Any combination of techniques that acted on the soft palate and oropharynx along with treatment of the base of the tongue or the epiglottis was possible. The variability in procedures, along with the inclusion of technical indications according to the availability at each hospital, may explain the success rate observed. In other hospitals, the surgical plan may not be performed according to DISE findings. Still, the use of modern combinations of techniques is key for successful multilevel surgery.

The significant improvements after surgery in this study are similar to those reported in two previous randomized clinical trials of surgeries conducted in patients with similar OSAHS severity [17,18]. However, these trials used only uvulopalatopharyngoplasty (UPPP) because of predominant retropalatal obstruction and/or palatine tonsil enlargement (i.e., Friedman stage I or II). Still, most patients with OSAHS have multilevel obstruction, including increased tongue size because of fat deposition [19]. Thus, the results of our study of multilevel surgery support a broader role for UA surgery to manage OSAHS.

The more extensive approach and multilevel surgery performed in one step that includes treatment at the oropharyngeal and hypopharyngeal levels of collapse used in this study had a larger treatment effect but a similar risk of adverse events as those of a large observational cohort treated predominantly with UPPP alone [20]. None of the patients in our study experienced serious adverse events potentially related to surgery or reported significant long-term functional difficulties.

Considering the results of a multicenter Australian trial [21] and that both nonanatomical and anatomical factors contribute to OSAHS [22], we anticipated that the surgical intervention would reduce but not eliminate obstructive breathing events in these patients.

The reduction in AHI was substantial and similar to oral appliances [23] and hypoglossal nerve stimulation [24]. Most importantly, we found substantial improvements in patient-centered outcomes in patients unable to use conventional OSAHS treatment. This surgery does not preclude the reintroduction of CPAP or other therapies later if required. The perception that CPAP treatment is problematic after pharyngoplasty arises from a single early report of more mouth leaks with nasal CPAP after excisional UPPP [25]. However, the modified pharyngoplasty used in this study reduces retropalatal obstruction while preserving the palate and velopharyngeal sphincter function, avoiding complications with CPAP usage if needed [26].

The use of alternative energy sources, such as those used in coblation, to remove the lingual tonsils has been shown to be noninferior to TORS multilevel surgery [6]. In addition, CTBR addresses the important issue of cost and accessibility. Endoscopic coblation at low temperatures offers some benefits over these methods: good visualization of the tissue using angulated endoscopes, little or no intraoperative bleeding, reduced swelling, and less postoperative pain [6,7].

Recent studies have reported that the surgical plan based on examination of awake patients changed after DISE in 40–50% of patients [27,28]. Improving success rates of sleep surgery [29,30] affects decision making [31] especially if the tongue was involved; however, there was a lack of evidence regarding the superiority of DISE over Müller’s maneuver with respect to surgical outcomes.

The strengths of this study include the follow-up to ascertain the effects of treatment independent of the short-term postoperative discomfort and low rates of participant withdrawal and loss to follow-up. One limitation of our study is that, although our study was adequately powered to establish efficacy, the generalizability from any study is inherently limited by the number of patients. The small sample size may limit generalization from our results. Further studies are needed to establish this surgical treatment’s long-term effectiveness, safety, and cost-effectiveness for OSAHS. Another limitation is that our study included a select population that excluded patients with severe obesity (BMI > 35 kg/m^2^), those older than 70 years, or those with retrognathia and significant comorbidities. In addition, women were underrepresented in this trial. Therefore, the results may not be generalizable to the larger OSAHS population.

## 5. Conclusions

The findings of this study suggest that multilevel surgery is a safe and successful procedure for the treatment of severe OSAHS. Multilevel surgery seems appropriate for patients with OSAHS whose treatment is not tolerable or as first-line treatment in selected patients with well-defined airway obstruction, based on the detection of UA collapses using DISE. Multilevel surgery in one step seems to help reduce the risk of UA collapse in younger, nonobese patients with moderate to severe OSAHS.

## Figures and Tables

**Figure 1 jcm-10-04822-f001:**
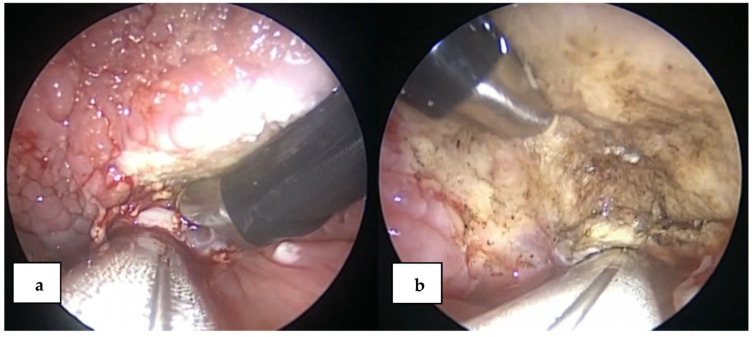
Exposure of the lingual tonsils: (**a**) surgical view before resection with coblation; (**b**) surgical view after resection with coblation in the same patient as in Figure 1a. Left-side resection was performed as well.

**Figure 2 jcm-10-04822-f002:**
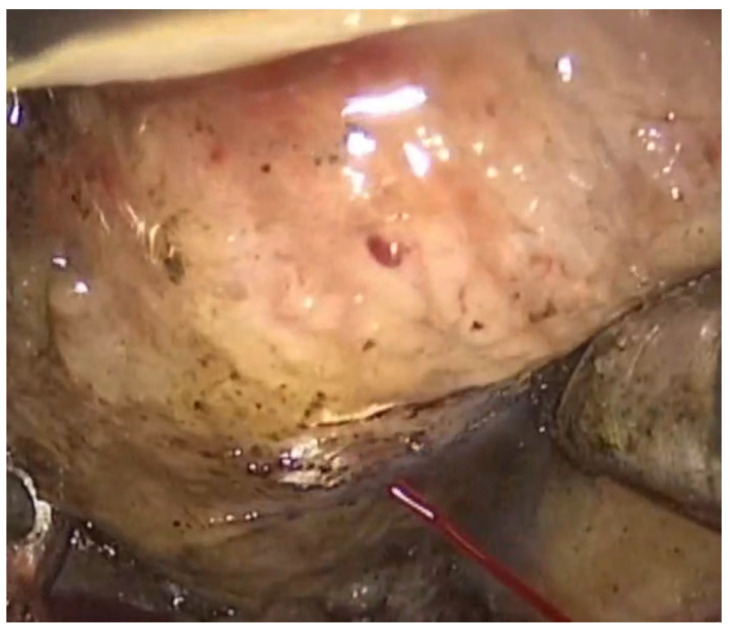
Intraoperative arterial bleeding during coblation, which probably originated in the left lingual artery and required electrocautery.

**Figure 3 jcm-10-04822-f003:**
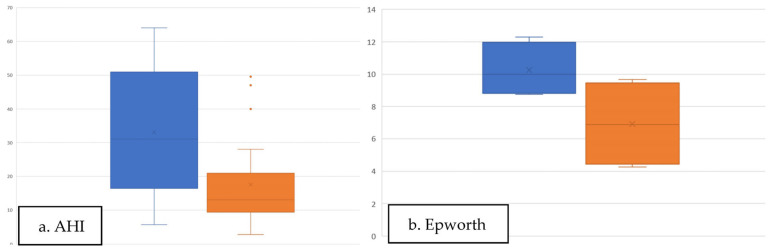
(**a**) Global mean AHI values before and after surgery; (**b**) Epworth Sleepiness Score before and after surgery.

**Figure 4 jcm-10-04822-f004:**
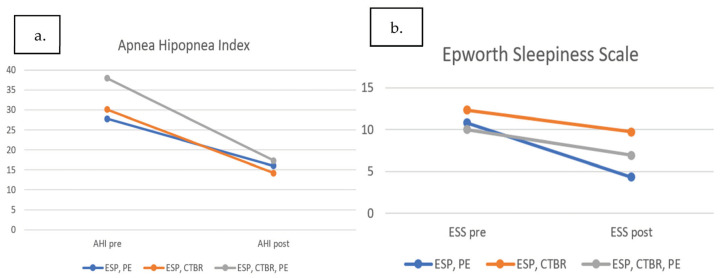
(**a**) Mean AHI values before and after surgery in the three surgical subgroups; (**b**) ESS before and after surgery in the three surgical subgroups: ESP, PE: expansion sphincter pharyngoplasty and partial epiglottectomy; ESP, CTBR: expansion sphincter pharyngoplasty and coblation tongue base reduction; ESP, CTBR, PE: expansion sphincter pharyngoplasty, coblation tongue base reduction, and partial epiglottectomy.

**Table 1 jcm-10-04822-t001:** Study demographics and clinical data.

Patient Characteristics	Mean ± Standard Deviation or Number (Percentage)
Age, years	49.16 ± 9.07
Gender, men/women	22/2
BMI, kg/m^2^	27.26 ± 2.78
ESS score	11 ± 5.11
AHI, events/h	33.01 ± 17.53
MinO_2_Sat	82.4% ± 10.30
TSat90	3.2
LTH	2.41 ± 0.97
Friedman stage I	2 (9%)
Friedman stage II	16 (66%)
Friedman stage III	6 (25%)
Müller’s maneuver	
Positive retropalatal collapse	10 (41%)
Positive retrolingual collapse	14 (60%)

BMI: body mass index, ESS: Epworth Sleepiness Scale, AHI: apnea–hypopnea index, MinO_2_Sat: lowest oxygen saturation, LTH: lingual tonsil hypertrophy grade, TSat90: time percentage with SatO_2_ < 90%.

**Table 2 jcm-10-04822-t002:** Results of VOTE classification during DISE.

Type of Surgery	V	O	T	E
ESP, PE	1	2	0	2
ESP, CTBR	1	2	2	0
ESP, CTBR, PE	1	2	2	2

ESP: expansion sphincter pharyngoplasty, PE: partial epiglottectomy; CTBR: coblation tongue base reduction.

**Table 3 jcm-10-04822-t003:** Responses of AHI and ESS score, and success rates for the three types of surgery.

Type of Surgery	Number of Patients	ESS Pre	ESS Post	AHI Pre	AHI Post	Sher’s Criteria	*p* Value
ESP, PE	6	10.8	4.3	27.8	16	No	>0.06
ESP, CTBR	10	12.3	9.7	30.1	14.2	Yes	<0.0001
ESP, CTBR, PE	8	10	6.9	37.9	17.3	Yes	<0.0001

ESS pre: preoperative Epworth Sleepiness Scale; ESS post: postoperative Epworth Sleepiness Scale; AHI pre: preoperative apnea–hypopnea index; AHI post: postoperative apnea–hypopnea index; ESP: expansion sphincter pharyngoplasty, PE: partial epiglottectomy; CTBR: coblation tongue base reduction; Sher’s criteria: postoperative AHI < 20 and/or 50% reduction in preoperative AHI.

## Data Availability

Data are contained within the article.

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
