# Peer review of "One-Stage Multilevel Surgery for Treatment of Obstructive Sleep Apnea Syndrome"

_jcm, 2021, doi:10.3390/jcm10214822_

Round 1
Reviewer 1 Report
Comments to the Author
I appreciate the authors' work in trying to define the one-stage multilevel surgery for the treatment of obstructive sleep apnea syndrome. I think that the paper presents some good data although the small patient number. However, there are still some issues that need to be addressed.
Relevance to mission: The information in this manuscript as it stands is interesting and may advance the current knowledge.
Internal validity: It seems that the study design, conduct, and analysis are described in a manner that is appropriate and reproducible.
External validity: The selection of included patients seems to be described in some detail.
Level of evidence: This manuscript is a retrospective descriptive study that may add eventually to the knowledge base beyond what is already published on this topic.
Title: “one-stage multilevel surgery for treatment of obstructive sleep apnea syndrome”
Conflicts: None affiliated with research reported.
Ethical conduct: The manuscript is original. There was mention of IRB approval. There is no mention of the conflict of interest.
Positive:
- Interesting topic.
- Well-written.
Concerns:
I will address some of the issues below
General:
The authors do not mention a hypothesis. As with any scientific paper, a clearly stated hypothesis helps to frame the manuscript for the reader.
METHODS:
- Please clarify the definition of “severe OSAS”. According to the guideline, AHI >30 represents the definition of “severe OSAS”. However, the author defined their” severe OSAS” as AHI>20.
- Please clarify the timing of “nasal surgery” in your patient group. When and how to perform your nasal surgery? Why not perform the nasal and palatal surgery at the same time?
- How to perform partial removal of the superior 1/3 of the epiglottis? Please describe in detail.
- Please compare the efficacy and safety regarding the one-stage multi-level surgery and multi-stage one-level surgery.
RESULTS:
- What were the adverse events in ESP, CTBR, and PE? Please clarify.
Readers may concern about the complication after one time multi-level surgery.
- The author didn’t show the results of VOTE classification during DISE. Please list it as a table.
- What is the association between Friedman classifications, Müller’s maneuver, and VOTE in your patient group?
- How to design your surgical plan according to Friedman classifications, Müller’s maneuver, and VOTE?
Author Response
Comments to the Editorial Board:
As suggested, the following changes have been made:
Reviewer #1
GENERAL:
“The authors do not mention a hypothesis. As with any scientific paper, a clearly stated hypothesis helps to frame the manuscript for the reader.”
As stated in lines 69-71, The hypothesis of our study would be in patients with moderately severe OSA, multilevel collapse of the UA confirmed in DISE and intolerance to CPAP, multilevel surgery can be considered as a treatment in one stage using coblation.
METHODS:
“Please clarify the definition of “severe OSAS”. According to the guideline, AHI >30 represents the definition of “severe OSAS”. However, the author defined their” severe OSAS” as AHI>20”.
We would like to thank the reviewer for this thoughtful review on our manuscript.
Lines 79-80.- Changed: a diagnosis of moderate OSAHS as defined by an apnea–hypopnea index (AHI) 5-29 and severe if AHI>30.
“Please clarify the timing of “nasal surgery” in your patient group. When and how to perform your nasal surgery? Why not perform the nasal and palatal surgery at the same time?”
Septoplasty was performed according to a modified Cottle’s technique with the use of a photophore, respecting the right upper superior tunnel without dissecting. The surgical technique was standardized and included a hemitransfixion incision followed by an elevation of the septal mucoperichondrium in both sides, addressing all areas of deviation and reshaping or removing the deviated part of the cartilage. Associated turbinoplasty with radiofrequency probe (TGH Endoscopia. BM-780 II. Radiofrequency Unit, Spain) under endoscopic control with a 0 degree optic was also performed as required. The radiofrequency probe was inserted into the tail, body and head of the inferior turbinate, at a fixed power of 2, during 8 seconds.
In another study published by the same author (Bosco G, Pérez-Martín N, Morato M, Racionero MA, Plaza G. Nasal surgery can improve upper airway collapse in patients with obstructive sleep apnea: an endoscopy study of drug-induced sleep. J Craniofac Surg. 2020 Jan / Feb; 31 (1): 68-71. Doi 10.1097 / SCS.0000000000005865. PMID: 31469731, we conducted a prospective evaluation of 34 patients with OSAHS, which underwent DISE before and after nasal surgery, with the aim to evaluate the effects of this procedure in the collapse of the UA. This study suggests that nasal surgery may improve hypopharyngeal collapses observed during DISE in patients with OSAHS. Thus, an improvement in nasal obstruction may also modify the surgical plan based on UA functional findings in OSAHS patients. According to this, nasal surgery is usually performed before palate and tongue surgery.
“How to perform partial removal of the superior 1/3 of the epiglottis? Please describe in detail.”
Lines 131-135.- Changed: The technique consists of a partial removal of 1/3 superior of the epiglottis in patients, mainly in patients with long, flaccid or trapdoor epiglottis that produced obstruction during DISE. The dissection through the coblation terminal is performed in a “V” shape. This protects the airway from aspiration. In addition, in case of presenting lateral collapse of the epiglottis, it can be resolved by this procedure.
“Please compare the efficacy and safety regarding the one-stage multi-level surgery and multi-stage one-level surgery”.
Data on multilevel surgery are very difficult to compare due to different patient selection criteria and operative techniques. So far, it is impossible to detect the more suitable combination of surgical techniques. Each surgeon must decide based on the characteristics of the patient, DISE, severity of OSA which combination works best in their hands. Regarding safety, there is an equal number of complications when comparing multilevel surgery with unilevel surgery. The key is in the selection of the patient and the experience of the surgical team.
RESULTS:
“What were the adverse events in ESP, CTBR, and PE? Please clarify. Readers may concern about the complication after one time multi-level surgery”.
Lines 222-223.- Changed: We had no major complications, bleeding or edema, no need for an urgent operating room. Only expected adverse events such as dysphagia resolved in the first month.
“The author didn’t show the results of VOTE classification during DISE. Please list it as a table”.
|
Type of surgery |
V |
O |
T |
E |
|
ESP, PE |
1 |
2 |
0 |
2 |
|
ESP, CTBR |
1 |
2 |
2 |
0 |
|
ESP, CTBR, PE |
1 |
2 |
2 |
2 |
“What is the association between Friedman classifications, Müller’s maneuver, and VOTE in your patient group?”.
Lines 231-236.- Changed: There is an association between Friedman and VOTE classifications in our sample of patients. Awake endoscopy determines that when there is severe lingual tonsil hypertrophy and the Friedman stage, there is a level predictor of collapse at the retrolingual level, showing a significant correlation with VOTE only when there is a severe retrolingual collapse. Friedman ́s stage is more correlated with VOTE with respect to retrolingual obstruction than the Müller’s maneuver.
“How to design your surgical plan according to Friedman classifications, Müller’s maneuver, and VOTE?”
As previously indicated, our ideal patient for multilevel surgery is one who shows 3-4 lingual tonsil hypertrophy, positive retrolingual Müller’s maneuver and lingual tongue hypertrophy and epiglottis fall as findings in the VOTE. According to our results, there can be no correlation with the Müller’s maneuver, therefore we do not take this last maneuver as an indication for this type of surgery.

Reviewer 2 Report
Dear Editor
and
Dear Authors
This is a very well-oriented study on the surgical techniques for the relief of OSAS symptoms. The strengths of the study are:
1) the use of the drug-induced sleep endoscopy (DISE) technique for the evaluation of the patients,
2) the department’s combination of techniques that facilitated concurrent intervention on the soft palate and oropharynx along with treatment of the base of the tongue or the epiglottis.
However, there are methodological gaps that must be clarified.
- Lines 79-80. “a diagnosis of severe OSAHS as defined by an apnea–hypopnea index (AHI) >20)”.
According to the American Academy of Sleep Medicine (AASM) criteria, patients with severe OSA have AHI>30/h. To be methodologically accurate, the authors have to classify patients into those with moderate OSA (AHI 15-30/h) and severe (AHI>30/h). Therefore, this statement needs to be rewritten. The statistical analysis, however, should be performed both as total and as separate.
- Lines 138-9. “The success of the surgery 138 was defined according to Sher´s criteria (Sleep 1996)”.
These criteria have raised many controversial arguments and tend not to be used as a postoperative evaluation (see: Caples SM, Rowley JA, Surgical modifications of the upper airway for obstructive sleep apnea in adults: a systematic review and meta-analysis. Sleep. 2010 Oct;33(10):1396-407. doi:10.1093/sleep/33.10.1396 and Aurora RN, et al; American Academy of Sleep Medicine. Practice parameters for the surgical modifications of the upper airway for obstructive sleep apnea in adults. Sleep. 2010 Oct;33(10):1408-13. doi: 10.1093/sleep/33.10.1408.). However, since in the literature there is an article that utilized these criteria (Lin HC etal. The efficacy of multilevel surgery of the upper airway in adults with obstructive sleep apnea/hypopnea syndrome Laryngoscope. 2008 May;118(5):902-8. doi: 10.1097/MLG.0b013e31816422ea), I suggest to accept this categorization, but only after re-categorization of the patients based on the most recent article of the same author that was used as a referee by the authors (Sher AE. Upper airway surgery for obstructive sleep apnea. Sleep Med Rev. 2002 Jun;6(3):195-212. doi: 10.1053/smrv.2002.0242) – presented in table 3 “Criteria defining surgical success”. Therefore, the most appropriate criteria would be “a postoperative AHI <15 and/or 50% reduction in preoperative AHI”, which also makes clinical sense – the patients will not need CPAP therapy.
- Lines 216-7. Sixteen patients (66%) showed a reduction in AHI >50% after surgery, and 75% of patients had an AHI value <20 postoperatively”.
Based on the above, 6 patients, amounting to 25% of all, did not show any improvement postoperatively. What are their specific characteristics (if any), which would suggest a possible mechanism for the failure of the procedure? Did they undergo the same procedure? These data have to be included in the manuscript.
- Lines 240-1. “arterial oxygen saturation measures (minO2Sat and SatO290), partner-reported snoring, and patient-reported sleep-specific quality of life.”
These data are nowhere displayed, not even in the supplementary files. It is suggested to include them in a separate column in Table 1.
- Line 241. “patient-reported sleep-specific quality of life”
How exactly was this measured, as there is no such clarification in the Methods section? Did the authors use a validate questionnaire?
- Line 270. “The reduction in AHI was substantial and similar to that for CPAP treatment (23)”
This reference (Ravesloot, MJL. deVries, N. Sleep 2011) showed that each patient’s time for using CPAP accords with the severity of the disease (AHI) and the therapeutic result is always an AHI<5. Thus, the comparison that the authors suggested with postoperative results is not appropriate.
- Line 304. “patients with OSAHS whose treatment is not effective”
Based on the results that were presented in the selection method of the patients, this sentence must be rephrased as “patients with OSAHS whose treatment is not tolerable”
- Lines 306-7. “Multilevel surgery in one step seems to help reduce the risk of UA collapse in patients with severe OSAHS.”
The exact conclusion of the manuscript would be more accurate with a statement to “younger, non-obese patients with moderate to severe OSAHS”.
Author Response
Comments to the Editorial Board:
As suggested, the following changes have been made:
Reviewer #2
Lines 79-80. “a diagnosis of severe OSAHS as defined by an apnea–hypopnea index (AHI) >20)”.
According to the American Academy of Sleep Medicine (AASM) criteria, patients with severe OSA have AHI>30/h. To be methodologically accurate, the authors have to classify patients into those with moderate OSA (AHI 15-30/h) and severe (AHI>30/h). Therefore, this statement needs to be rewritten. The statistical analysis, however, should be performed both as total and as separate.
We would like to thank the reviewer for this thoughtful review on our manuscript.
Lines 79-80. - Changed: a diagnosis of moderate OSAHS as defined by an apnea–hypopnea index (AHI) 5-29 and severe if AHI>30.
Lines 138-9. “The success of the surgery was defined according to Sher´s criteria (Sleep 1996)”.
These criteria have raised many controversial arguments and tend not to be used as a postoperative evaluation (see: Caples SM, Rowley JA, Surgical modifications of the upper airway for obstructive sleep apnea in adults: a systematic review and meta-analysis. Sleep. 2010 Oct;33(10):1396-407. doi:10.1093/sleep/33.10.1396 and Aurora RN, et al; American Academy of Sleep Medicine. Practice parameters for the surgical modifications of the upper airway for obstructive sleep apnea in adults. Sleep. 2010 Oct;33(10):1408-13. doi: 10.1093/sleep/33.10.1408.). However, since in the literature there is an article that utilized these criteria (Lin HC etal. The efficacy of multilevel surgery of the upper airway in adults with obstructive sleep apnea/hypopnea syndrome Laryngoscope. 2008 May;118(5):902-8. doi: 10.1097/MLG.0b013e31816422ea), I suggest to accept this categorization, but only after re-categorization of the patients based on the most recent article of the same author that was used as a referee by the authors (Sher AE. Upper airway surgery for obstructive sleep apnea. Sleep Med Rev. 2002 Jun;6(3):195-212. doi: 10.1053/smrv.2002.0242) – presented in table 3 “Criteria defining surgical success”. Therefore, the most appropriate criteria would be “a postoperative AHI <15 and/or 50% reduction in preoperative AHI”, which also makes clinical sense – the patients will not need CPAP therapy.
We appreciate the reviewer comments and feedback on this work.
Lines 143-4: Changed: The success of the surgery was defined according to Sher´s criteria (Sleep Med Rev 2002) [14] as a postoperative AHI <15 and/or 50% reduction in preoperative AHI.
Lines 245.– Changed in table 3: Sher`s criteria "no", instead of yes and, p value > 0.06.
Lines 228-9.– Changed: According to Sher’s criteria, the surgical success rate was significant in two subgroups.
Lines 216-7. Sixteen patients (66%) showed a reduction in AHI >50% after surgery, and 75% of patients had an AHI value <20 postoperatively”.
Based on the above, 6 patients, amounting to 25% of all, did not show any improvement postoperatively. What are their specific characteristics (if any), which would suggest a possible mechanism for the failure of the procedure? Did they undergo the same procedure? These data have to be included in the manuscript.
The patients were treated with multilevel surgery following individualized criteria according to the DISE findings, as explained in materials and methods. All patients underwent the same procedure within their subgroup. The subgroup that does not meet the success criteria is the one that undergoes ESP and PE only. The subgroups that meet the success criteria are those that undergo ESP and CTBR, or ESP, CTBR and PE. Therefore, using Coblation in the tongue base seems to be the successful key.
Lines 240-1. “arterial oxygen saturation measures (minO2Sat and SatO290), partner-reported snoring, and patient-reported sleep-specific quality of life.”
These data are nowhere displayed, not even in the supplementary files. It is suggested to include them in a separate column in Table 1.
We would like to thank the reviewer for this thoughtful comment and constructive suggestions.
The minO2Sat is included in Table 1. The SatO290 is now added to Table 1. Partner-reported snoring, and patient-reported sleep-specific quality of life are variables that the authors want to publish in another study in the longer term, so it is preferred not to include the data in this article.
Line 178.- Changed: SatO290 was 3.2.
Line 241. “patient-reported sleep-specific quality of life”
How exactly was this measured, as there is no such clarification in the Methods section? Did the authors use a validate questionnaire?
We appreciate this thoughtful comment from the reviewer.
The Pittsburgh Sleep Quality Index (PSQI) was validated questionnaire. Clinical properties of the PSQI were assessed over 23.3 months (range 12–36) period with "good” sleepers and "poor" sleepers.
Line 270. “The reduction in AHI was substantial and similar to that for CPAP treatment (23)”
This reference (Ravesloot, MJL. deVries, N. Sleep 2011) showed that each patient’s time for using CPAP accords with the severity of the disease (AHI) and the therapeutic result is always an AHI<5. Thus, the comparison that the authors suggested with postoperative results is not appropriate.
Following the reviewer's recommendations, part of the sentence is removed. So that finally remains:
Lines 288.- Changed: The reduction in AHI was substantial and similar to oral appliances.
Line 304. “patients with OSAHS whose treatment is not effective”
Based on the results that were presented in the selection method of the patients, this sentence must be rephrased as “patients with OSAHS whose treatment is not tolerable”
Lines 323.- Changed: patients with OSAHS whose treatment is not tolerable.
Lines 306-7. “Multilevel surgery in one step seems to help reduce the risk of UA collapse in patients with severe OSAHS.”
The exact conclusion of the manuscript would be more accurate with a statement to “younger, non-obese patients with moderate to severe OSAHS”.
Lines 325-6.- Changed: Multilevel surgery in one step seems to help reduce the risk of UA collapse in younger, non-obese patients with moderate to severe OSAHS.

Round 2
Reviewer 1 Report
Thank you to the authors for addressing my comments. This manuscript reads well and contains pertinent data.
Reviewer 2 Report
The author made all corrections needed.